# Evaluation of Blood Levels of Omentin-1 and Orexin-A in Adults with Obstructive Sleep Apnea: A Systematic Review and Meta-Analysis

**DOI:** 10.3390/life13010245

**Published:** 2023-01-16

**Authors:** Iman Mohammadi, Masoud Sadeghi, Golnaz Tajmiri, Annette Beatrix Brühl, Laleh Sadeghi Bahmani, Serge Brand

**Affiliations:** 1Department of Oral and Maxillofacial Surgery, Dental Implants Research Center, Dental Research Institute, School of Dentistry, Isfahan University of Medical Sciences, Isfahan 8174673461, Iran; 2Department of Biology, Science and Research Branch, Islamic Azad University, Tehran 1477893855, Iran; 3Dental Implants Research Center, Dental Research Institute, School of Dentistry, Isfahan University of Medical Sciences, Isfahan 8174673461, Iran; 4Center for Affective, Stress and Sleep Disorders (ZASS), Psychiatric University Hospital Basel, 4002 Basel, Switzerland; 5Department of Education and Psychology, Shahid Ashrafi Esfahani University, Ishafan 8179949999, Iran; 6Sleep Disorders Research Center, Kermanshah University of Medical Sciences, Kermanshah 6715847141, Iran; 7Department of Sport, Exercise and Health, Division of Sport Science and Psychosocial Health, University of Basel, 4052 Basel, Switzerland; 8Substance Abuse Prevention Research Center, Kermanshah University of Medical Sciences, Kermanshah 67146, Iran; 9School of Medicine, Tehran University of Medical Sciences, Tehran 25529, Iran; 10Center for Disaster Psychiatry and Disaster Psychology, Psychiatric University Hospital Basel, 4002 Basel, Switzerland

**Keywords:** omentin protein, orexins, serum, plasma, meta-analysis

## Abstract

Background and objective: Obstructive sleep apnea (OSA) can be related to changes in the levels of adipokines and neuropeptides, which in turn may affect the energy balance components of neuronal cells. Herein, a systematic review and meta-analysis checked the changes in serum/plasma levels of omentin-1 (OM-1: an adipokine) and orexin-A (OXA: a neuropeptide) in adults (age > 18 years old) with OSA (aOSA) compared to controls. Materials and methods: Four databases (Cochrane Library, PubMed, Web of Science, and Scopus) were systematically searched until 14 November 2022, without any restrictions. The Joanna Briggs Institute (JBI) critical appraisal checklist adapted for case–control studies was used to assess the quality of the papers. The effect sizes were extracted using the Review Manager 5.3 software for the blood levels of OM-1 and OXA in aOSA compared with controls. Results: Thirteen articles, with six studies for OM-1 levels and eight for OXA levels, were included. The pooled standardized mean differences were −0.85 (95% confidence interval (CI): −2.19, 0.48; *p* = 0.21; I^2^ = 98%) and −0.20 (95%CI: −1.16, 0.76; *p* = 0.68; I^2^ = 96%) for OM-1 and OXA levels, respectively. Among the studies reporting OM-1, five were high and one was moderate quality. Among the studies reporting OXA, six were moderate, one was high, and one was low quality. Based on the trial sequential analysis, more participants are needed to confirm the pooled results of the analyses of blood levels of OM-1 and OXA. In addition, the radial plot showed outliers as significant factors for high heterogeneity. Conclusions: The main findings indicated a lack of association between the blood levels of OM-1 and OXA and OSA risk. Therefore, OM-1 and OXA did not appear to be suitable biomarkers for the diagnosis and development of OSA.

## 1. Introduction

Obstructive sleep apnea (OSA) is a status determined by repeated episodes of partial or complete airway obstruction during sleep [1,2], which usually remains undiagnosed and untreated [3]. A systematic review in 2017 reported that the overall prevalence of any OSA ranged from 9% to 38% in the general adult population, from 13% to 33% and 6% to 19% in females and males, respectively, while the overall prevalence increases in older adults [4]. The global prevalence of moderate to severe OSA is approximately 20% [5]. Clinically, the apnea-hypopnea index (AHI) provides well-established and useful cutoff-values which can be applied to diagnose both the OSA severity and to assess the effects of interventions [6,7]. A diagnosis using polysomnography (PSG: the gold standard test for diagnosing OSA) [8,9] with an AHI ≥ 5 is considered indicative of OSA among adults [8,10,11].

Several factors including environmental and genetic factors, and even internal conditions (hormones, cytokines, and other biomarkers), might lead to the emergence and maintenance of OSA. Smoking [12] and alcohol consumption [13], and alterations in the blood levels of intercellular adhesion molecule-1 [14], monocyte chemoattractant protein-1 [15], adiponectin [16], C-reactive protein [17], cortisol [18], interleukin-6 [19], tumor necrosis factor-alpha [20], astrocytic protein (S100B) [21], neuron-specific enolase [21], and polymorphisms [22,23,24] were observed among adults with OSA, compared to healthy controls.

Furthermore, OSA is linked to other diseases such as diabetes, hypertension, cardiovascular diseases, congestive heart failure, and stroke [25,26]; therefore, OSA can be seen as a risk factor for other diseases. As such, the evaluation and treatment of OSA should focus on the recognition of patients at risk of OSA [27]. Obesity is the highest risk factor for OSA and it is estimated that 58% of moderate-to-severe OSA development can be attributed to obesity [28]. Continuous positive airway pressure is still the main treatment for patients with OSA because of its effect on the symptoms and on quality of life [29,30].

Obesity—an energy balance disorder—is associated with a positive energy balance, understood as an imbalance between higher energy storage and energy consumption [31,32]. Orexin-A (OXA or hypocretin-1) is a neuropeptide that plays a role in appetite [33], sleep, and arousal arrangement [34]. OXA can play a role in obesity [35]. The adipocytokine omentin-1 (OM-1 or intelectin-1) is expressed in visceral adipose tissue and is related to the inflammatory response [36]. The reduced serum levels of OM-1 are linked with obesity and type 2 diabetes mellitus (T2DM) [37]. OM-1 may be necessary for holding the energy balance in a time of insulin resistance pathogenesis and T2DM [38]. The energy balance components can be affected by OSA and thus allow clinicians to more effectively guide overall therapeutic approaches to optimize weight loss and promote cardiovascular health in individuals with OSA [39].

A number of studies have evaluated the association between blood OM-1 [40,41] and OXA [42,43] levels in adults with OSA (aOSA) compared to controls, but there were different and, above all, contrasting results. In addition, based on our knowledge, there was no previous meta-analysis in the English-language literature. Therefore, we designed a systematic review and meta-analysis evaluating the association between blood OM-1 and OXA levels in aOSA versus controls. 

## 2. Materials and Methods

To design the present meta-analysis, the PRISMA-P items [44] were followed. The PECO question [45,46] was: Are blood OM-1 and OXA levels different in aOSA in comparison with controls? (P: human adults with/without OSA, E: OSA disorder, C: aOSA in comparison with controls; and O: the plasma/serum levels of OM-1 or OXA). 

### 2.1. Search Strategy 

The databases of Cochrane Library, PubMed, Web of Science, and Scopus were comprehensively searched by one author (M.S.) until 14 November 2022, without any restrictions. The search terms were: (“obstructive sleep apnea” or “OSA” or “sleep apnea” or “obstructive sleep apnea-hypopnea syndrome” or “OSAHS” or “obstructive sleep apnea syndrome” or “OSAS”) AND (“hypocretin” or “orexin-A” or “OXA” or “omentin-1” or “OM-1” or “intelectin-1”). The citations of any article linked to the subject and the databases of “Google Scholar”, “Elsevier publishers”, and “BMJ publishers” were checked so that no study was missed. Two authors working independently (M.S. and I.M.) were responsible for removing irrelevant articles and excluding articles with reasons. There was no disagreement.

### 2.2. Eligibility Criteria

Inclusion criteria: (1) Studies involving both groups of adults (aOSA and controls); (2) Both groups had no treatment during sampling; (3) In both groups, participants were 18 years old or older, with no restrictions on BMI values; (4) Studies reporting plasma/serum hypocretin, orexin-A, omentin-1, or interlectin-1 levels in OSA and controls; (5) PSG was used for diagnosing OSA; (6) OSA in adults was defined as an AHI ≥ 5 events/h; (7) Participants included in the studies did not report other systemic diseases (cardiovascular diseases; diabetes mellitus; infectious diseases; any malignancy; heart, renal, and hepatic failures; lung diseases; and other sleep disorders); (8) Controls had no OSA or systemic diseases (see the previous criterion); (9) Venous blood was taken in a fasting state in the morning to measure OM-1 and OXA levels. Exclusion criteria: (1) Meta-analyses, letters to the editor, book chapters, conference papers, reviews, and commentaries; (2) Studies with a lack of complete data; (3) Studies with no control group. 

### 2.3. Data Collection

The data were extracted for any study by one author (I.M.), including the country and ethnicity of participants, the first author, the publication year, OM-1 and OXA sampling, the sample size of both groups (aOSA and controls), mean BMI, mean age and mean AHI of two groups, quality or quality score, and mean serum/plasma levels of OM-1 and OXA in two groups. Another author (G.T) re-checked the process of extraction; in case of disagreements between the two authors’ decisions, a discussion between all three authors resolved the issue.

### 2.4. Quality Assessment

Two authors (M.S. and S.B.) independently evaluated the quality based on the Joanna Briggs Institute (JBI) critical appraisal checklist adapted for case–control studies including ten questions or ten scores. Low-quality studies had a score of 1–4; moderate- quality studies, 5–7; and high-quality studies: 8–10 [47] (See Appendix A). 

### 2.5. Statistical Analyses

The statistical analyses were performed by one author (M.S.). The standardized mean difference (SMD) and 95% confidence interval (CI)) of blood levels of OM-1 and OXA amongst aOSA and controls were extracted using the Review Manager 5.3 (RevMan 5.3; the Cochrane Collaboration, the Nordic Cochrane Centre, Copenhagen, Denmark) software. A significant value was considered when the *p*-value (2-sided) was less than 0.05. The significant heterogeneity was considered while P_heterogeneity_ < 0.1 (I^2^ > 50%) and in the state, the analysis was performed in a random-effect model [48]; otherwise, it was performed in a fixed-effect model [49]. In addition, the radial plot result was reported to confirm or reject the heterogeneity due to outliers among the studies. 

The subgroup and mixed-effect meta-regression analyses were based on several variables, and the stability of initial pooled SMDs was assessed by both “one-study-removed” and “cumulative” analyses as sensitivity analyses. 

Begg’s and Egger’s tests were applied to test the potential publication bias [50] and to report the degree of asymmetry [51], respectively; the data for publication bias and sensitivity analyses were extracted using the Comprehensive Meta-Analysis version 2.0 (CMA 2.0; Biostat Inc., Englewood, NJ, USA) software, with a *p*-value (2-sided) less than 0.10 recommending the existence of the significant publication bias. 

To report the potential random error (false-positive and -negative results) in the meta-analysis [52], trial sequential analysis (TSA) was accomplished using TSA software (version 0.9.5.10 beta) (Copenhagen Trial Unit, Centre for Clinical Intervention Research, Rigshospitalet, Copenhagen, Denmark) [53]. The futility threshold could show a no-impact result before attaining the information size. To calculate the required information size (RIS), an α-risk of 5%, β-risk of 20%, and two-sided border type were used to extract the mean difference and variance according to the empirical assumptions created by the TSA software automatically. If the Z-curve reached the RIS line, enough participants were included in the studies, the conclusions were reliable or crossed the boundaries, and the results could be strong. On the other hand, the volume of information was not large enough and more evidence was needed. 

Some studies reported the effect sizes based on median and 25th–75th percentiles [54] or median (min-max) [55], and in the state, we changed them to the mean (standard deviation) based on just entering the format into the software.

All authors checked the last version of the article and the disagreement was resolved by a discussion between them.

## 3. Results

### 3.1. Study Selection

To search the databases, 716 records were extracted, and after removing duplicates and irrelevant records, 26 full-text articles were assessed using the eligibility criteria. Then, 13 full-text articles were removed for different reasons (Figure 1). Finally, 13 articles including 6 studies reporting blood OM-1 levels and 9 reporting OXA levels entered the meta-analysis. 

### 3.2. Characteristics of Studies

Table 1 reports the data for the variables of fourteen studies from the thirteen articles [40,41,42,43,56,57,58,59,60,61,62,63,64] included in the meta-analysis. With regard to the studies reporting OM-1 levels, four were reported among Caucasians and two among Asians, and for the studies reporting OXA levels, three were among Caucasians and five among Asians. For OM-1, three studies reported plasma and three reported serum levels; however, for OXA, seven reported plasma levels and one reported the serum level.

### 3.3. Quality Assessment

Table 2 reports the JBI critical appraisal checklist for both blood OM-1 and OXA levels in case–control studies. Among the studies reporting OM-1, five were high and one was moderate quality. Among the studies reporting OXA, six were moderate, one was high, and one was low quality. The questions for the JBI critical appraisal checklist are located in Appendix A.

### 3.4. Meta-Analysis for OM-1 Levels

Figure 2 shows the random-effect analysis of blood OM-1 levels in aOSA compared to controls. The pooled result showed a lack of association between the blood levels of OM-1 and OSA risk (SMD = −0.85; 95%CI: −2.19, 0.48; *p* = 0.21; I^2^ = 98%). 

### 3.5. Meta-Analysis for OXA Levels

Figure 3 illustrates the random-effect analysis of blood OXA levels in aOSA compared to controls. The pooled result showed a lack of association between the blood levels of OXA and OSA risk (SMD = −0.20; 95%CI: −1.16, 0.76; *p* = 0.68; I^2^ = 96%). 

### 3.6. Subgroup Analysis for OM-1 Levels

Table 3 reports the subgroup analysis based on the variables for finding confounding factors in the pooled analysis of the serum/plasma levels of OM-1 in aOSA compared with controls. The results reported that the mean AHI of aOSA and quality were two confounding factors in the pooled SMD.

### 3.7. Subgroup Analysis for OXA Levels

Table 4 reports the subgroup analysis based on the variables for finding confounding factors in the pooled analysis of the serum/plasma levels of OXA in aOSA compared to controls. The results reported that the mean BMI in controls, the mean age of aOSA, and the quality were three confounding factors in the pooled SMD.

### 3.8. Meta-Regression Analysis

As Table 5 shows, among the eight variables checked in the meta-regression analysis for both OM-1 and OXA levels, just sample size was an effective factor for the pooled SMD of blood OM-1 levels (*p* = 0.00445). With an increasing sample size, OM-1 levels were significantly reduced. 

### 3.9. Radial Plot

Figure 4 shows radial plots for OM-1 and OXA levels in aOSA compared to controls. The results confirmed a significant high heterogeneity due to outliers among the studies reporting OM-1 (Cochran’s Q = 46.01; *p* < 0.01) and OXA (Cochran’s Q = 299.08; *p* < 0.01) levels.

### 3.10. Sensitivity Analysis

The sensitivity analyses (one-study-removed and cumulative analyses) showed the stability of the pooled results for both OM-1 and OXA levels in aOSA compared to controls. The plots for the sensitivity analyses are located in the Appendix A.

### 3.11. Trial Sequential Analysis (TSA)

Figure 5 shows the TSAs of the blood levels of OM-1 and OXA in aOSA compared to controls. The TSA of six studies for blood OM-1 levels shows the line of the cumulative Z-curve crossed the line of the trial sequential monitoring boundary for benefit, but not the line of RIS (heterogeneity (D^2^) = 98%). The result was robust, before reaching the information size needed to obtain reliable evidence adjusted for random error risk and suggesting further studies. The TSA of eight studies for blood OXA levels shows the line of the cumulative Z-curve did not cross any line (D^2^ = 99%). Therefore, it indicated an absence of evidence if the information size is not reached and that further studies are needed. 

### 3.12. Publication Bias

Figure 6 shows the funnel plots of serum/plasma OM-1 and OXA levels with the results of the trim-fill method in aOSA compared to controls. The p-values of Egger’s/ Begg’s tests for OM-1 and OXA levels were 0.516/0.851 and 0.161/0.458, respectively. For serum/plasma OM-1 levels and having one imputed study, under the fixed-effects model, the point estimate and pseudo 95% CI for the combined studies was −1.214 (−1.389, −1.039); using the trim-fill method, the imputed point estimate was −1.292 (−1.463, −1.120). In addition, under the random-effects model, the point estimate and 95% CI for the combined studies was −0.837 (−2.197, 0.523); using the trim-fill method, the imputed point estimate was −1.190 (−2.764, 0.085). Therefore, the overall effect sizes on blood OM-1 levels reported in the forest plot appeared valid; with a trivial publication bias effect based the on fixed-effects and random-effects models, because the observed estimates were similar to the adjusted estimates.

For serum/plasma OXA levels and without any imputed study, under the fixed-effects model, the point estimate and pseudo 95% CI for the combined studies was 0.307 (0.126, 0.487); using the trim-fill method, the imputed point estimate was 0.307 (0.126, 0.487). In addition, under the random-effects model, the point estimate and 95% CI for the combined studies was −0.211 (−1.189, 0.766); using the trim-fill method, the imputed point estimate was −0.211 (−1.189, 0.766). Therefore, the overall effect sizes on blood OXA levels reported in the forest plot appeared valid, with a trivial publication bias effect based on the fixed-effects and random-effects models, because the observed estimates were similar to the adjusted estimates.

## 4. Discussion

Despite a low number of cases and high heterogeneity, the results of the meta-analysis represented a lack of association between the blood levels of OM-1 and OXA and OSA risk. Sample size, mean AHI of aOSA, and quality were confounding factors for OM-1 levels, whereas, mean BMI in controls, mean age of aOSA, and quality were confounding factors for OXA levels. 

OSA must be investigated as a multifactorial disease in which multiple polymorphisms, environmental influences, and developmental factors are closely related to it [65]; OSA is also strongly correlated with obesity [66]. The main mechanisms of OSA are hypoxia and oxidative stress; however, several studies have shown that inflammation also plays an important role in the occurrence and progression of OSA [67,68,69]. OSA appears to affect parameters involved in the regulation of energy balance, including food consumption, physical activity, energy metabolism, and the hormonal regulation of hunger/satiety [70,71].

There is a relationship between OM-1 level, body composition, and physical activity in elderly women [72]. Aging, physical activity, and body weight are several significant factors affecting OM-1 concentration. However, it is not clear what factors can be used as predictors of the impact of the circulating OM-1 concentration [73,74]. OXA stimulates both nutrition and energy intake, and OXA’s increase in energy intake is mainly due to increased spontaneous physical activity, and this effect on energy intake is stronger than the effect on nutrition [75]. OXA has been demonstrated to cause obesity and its related diseases [35].

Three studies [41,60,62] reported that the OM-1 levels were significantly lower in aOSA than those in controls. The decreased levels of OM-1 are related to insulin resistance T2DM, coronary artery disease, and arterial stiffness, or on the other hand, associated inversely with metabolic syndrome [76,77,78,79,80,81]. In contrast, two studies [40,64], in line with the present meta-analysis, reported there was no association between OM-1 levels and OSA risk. These different results of the studies can be due to the differences in sample size, BMI status, age, and quality of the studies.

Three studies [42,59,63] showed a reduced significant level of OXA in aOSA compared to controls. In contrast, three articles [43,57,58] reported an elevated significant level of OXA in aOSA compared to controls. Low levels of OXA may prevent arousal during sleep or may be the result of sleepiness from repeated arousals caused by a mechanism independent of OXA [42], and also obesity in individuals with OSA may be induced by decreased orexin neuron activity or levels of OXA [82]. Higher levels of OXA were related to high breakfast energy intake [83]. In addition, elevated orexin transmission, reflected as elevated plasma levels of OXA, may impact the arousal response in individuals with OSA [57]. The results of the present meta-analysis are in line with another study [61] which found a lack of association between OXA levels in aOSA versus controls. These different results of the studies can be due to the differences in AHI status and the quality of the studies.

The present results should be counterbalanced by the following limitations: (1) A low number of cases in most studies; (2) A low number of studies in each analysis; (3) A high heterogeneity across or among the studies; (4) Some studies were outliers; (5) A lack of high quality in most studies. In contrast, there were two strengths: (1) Stability of the pooled results; (2) A lack of publication bias. 

## 5. Conclusions

The main results indicated a lack of association between the blood levels of OM-1 and OXA and OSA risk. Therefore, OM-1 and OXA did not appear to be suitable biomarkers for the diagnosis and development of OSA. Although we used eligibility criteria, our results must be interpreted with precaution, because the levels of the two biomarkers may be impacted by several factors (age, BMI, AHI, sample size, etc.). Finally, the sample sizes in most studies were relatively small, and, therefore, larger and multi-center studies should be analyzed to clarify the relationship between the blood levels of these biomarkers and OSA development.

## Figures and Tables

**Figure 1 life-13-00245-f001:**
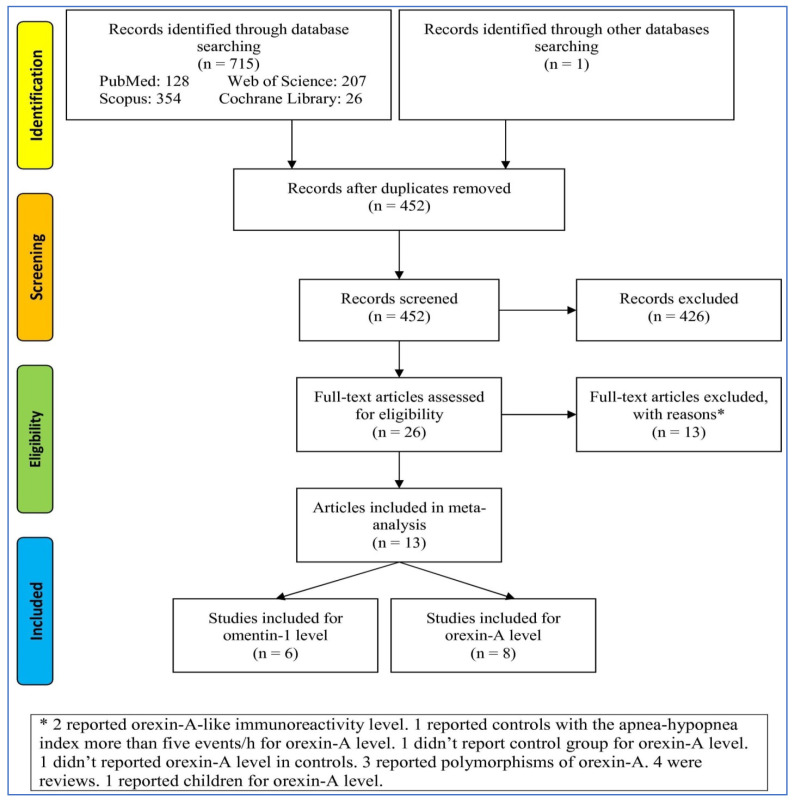
Flowchart of the study selection. * The reasons of exclusion of full-text articles.

**Figure 2 life-13-00245-f002:**
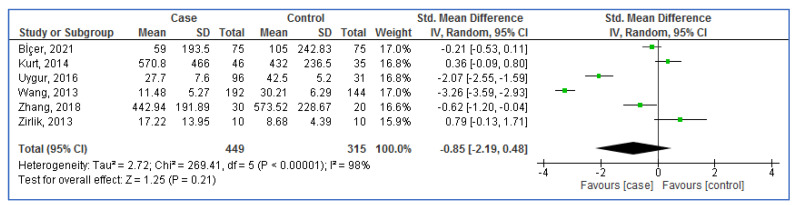
Forest plot analysis of blood omentin-1 levels in adults with obstructive sleep apnea compared to controls. SD: Standard deviation. CI: Confidence interval.

**Figure 3 life-13-00245-f003:**
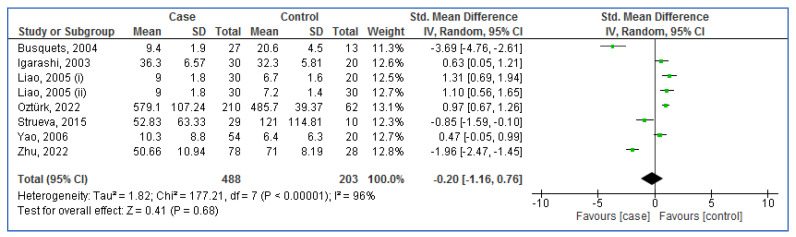
Forest plot analysis of blood orexin-A levels in adults with obstructive sleep apnea compared to controls. SD: Standard deviation. CI: Confidence interval.

**Figure 4 life-13-00245-f004:**
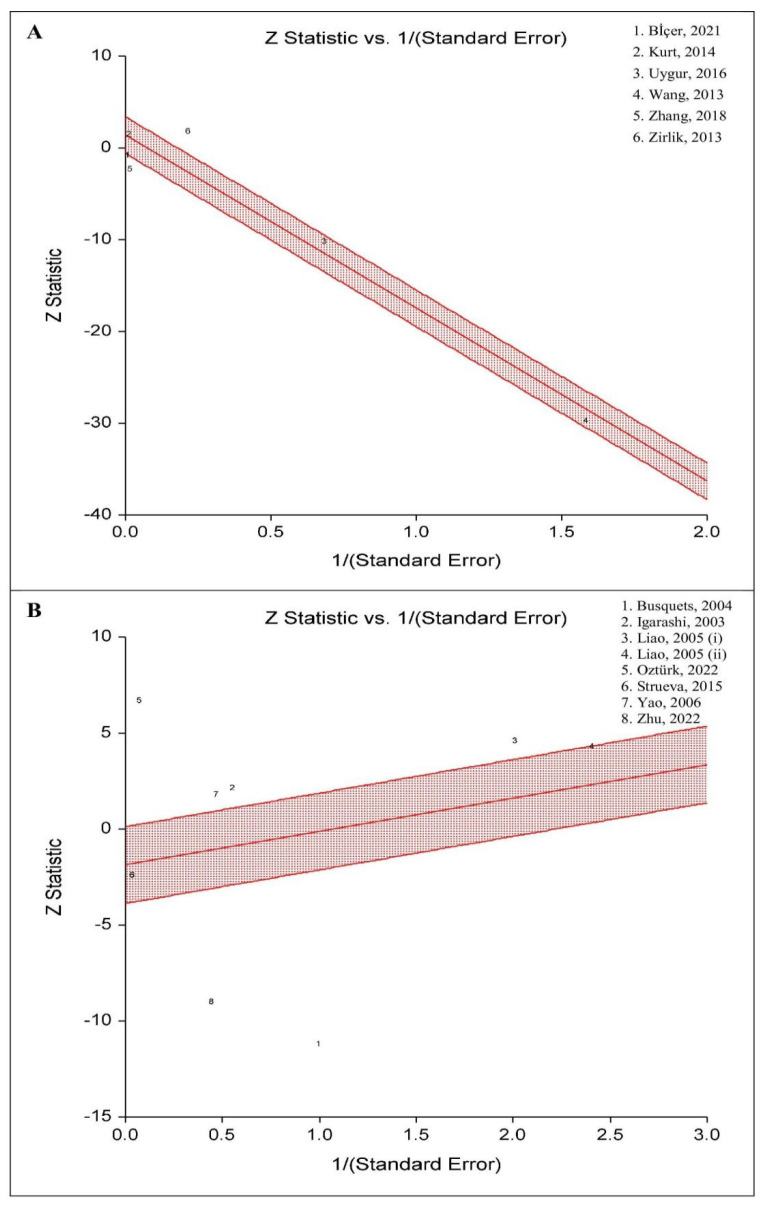
The radial (or Galbraith) plot of the blood levels of (**A**) omentin-1 and (**B**) orexin-A in adults with obstructive sleep apnea compared to controls. This plot shows the z-statistic on the Y axis and 1/standard error on the X axis. Studies within and outside the limits are interpreted as homogeneous and may be outliers, respectively.

**Figure 5 life-13-00245-f005:**
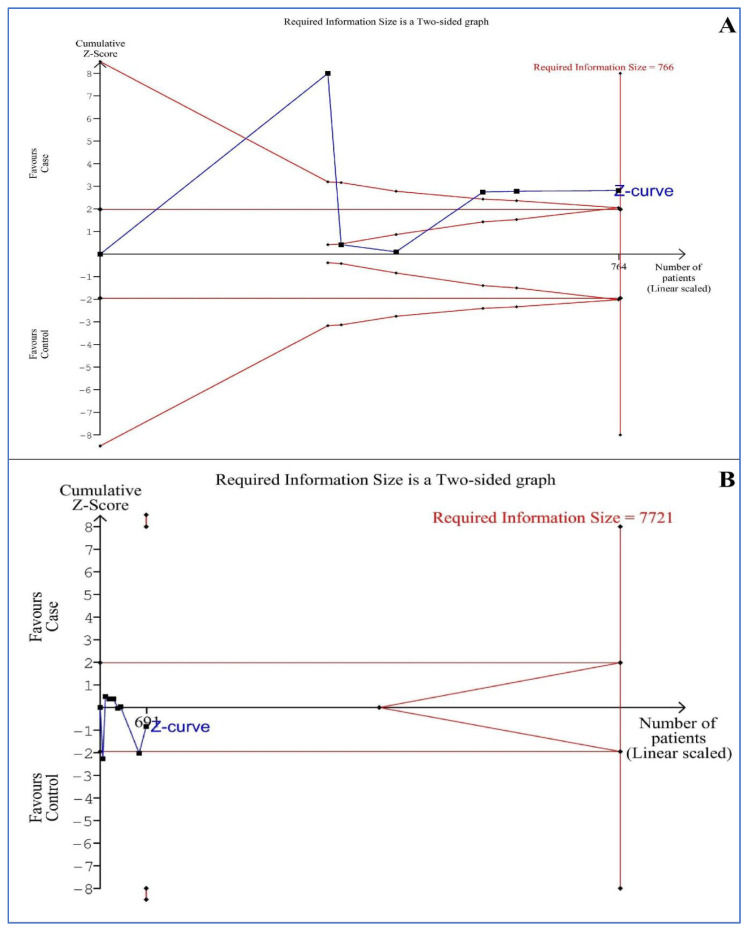
The trial sequential analyses of the blood levels of (**A**) omentin-1 and (**B**) orexin-A in adults with obstructive sleep apnea compared to controls. Each black square fill icon shows a study. X-axis and Y-axis show the number of patients and cumulative Z-score, respectively. Horizontal brown lines: conventional boundaries for benefit (up) or harm (down). Sloping full red lines with black square fill icons: trial sequential monitoring boundaries for benefit (up) or harm (down), and futility boundaries. Full blue line with black square fill icons: Z-curve. Vertical red full line: Required information size boundary.

**Figure 6 life-13-00245-f006:**
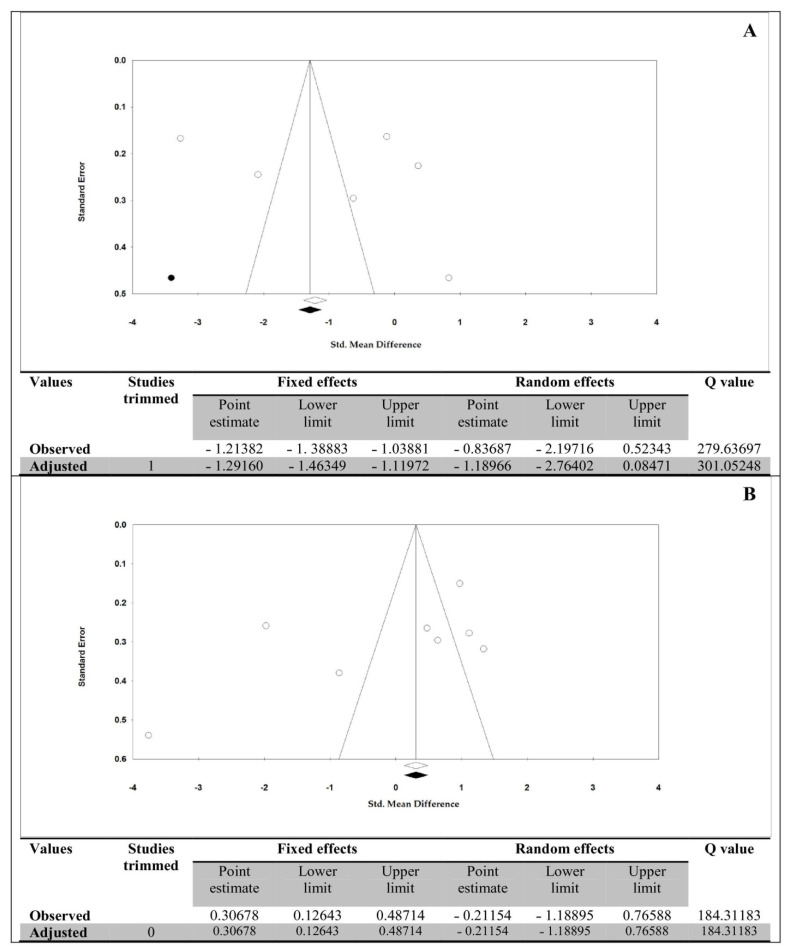
Funnel plots of the blood levels of (**A**) omentin-1 and (**B**) orexin-A in adults with obstructive sleep apnea compared to controls with the trim-fill method. Each white circle shows an included study in the meta-analysis and each black circle shows an unpublished study. The white diamond shows overall effect size as calculated in the meta-analysis, and the black diamond shows the corrected effect size.

**Table 1 life-13-00245-t001:** Characteristics of the studies.

First Author, Publication Year	Country	Ethnicity	Case				Control				Sample	Measured Factor
			Number	Mean Age, Years	Mean BMI, kg/m^2^	Mean AHI, events/h	Number	Mean Age, Years	Mean BMI, kg/m^2^	Mean AHI, events/h		
Igarashi, 2003 [57]	Japan	Asian	30	45.3	28.6	31.8	20	46.5	26.4	1.9	Plasma	Orexin-A
Busquets, 2004 [42]	Spain	Caucasian	27	52	31	54	13	46	24	<5	Plasma	Orexin-A
Liao, 2005 (i) [43]	China	Asian	30	53	30.2	≥5	20	48	22.1	1.7	Plasma	Orexin-A
Liao, 2005 (ii) [43]	China	Asian	30	53	30.2	≥5	30	51	30.9	2.3	Plasma	Orexin-A
Yao, 2006 [61]	China	Asian	54	47	34	64	20	45	32	3.5	Plasma	Orexin-A
Wang, 2013 [41]	China	Asian	192	49.2	26.7	24	144	48.7	27.1	2	Serum	Omentin-1
Zirlik, 2013 [64]	Germany	Caucasian	10	58.9	31.7	40.4	10	53.6	26.7	2.7	Plasma	Omentin-1
Kurt, 2014 [40]	Turkey	Caucasian	46	48.1	30.4	≥5	35	42.8	26.4	<5	Serum	Omentin-1
Strueva, 2015 [59]	Russia	Caucasian	29	41	39.2	≥30	10	38	22.4	<5	Plasma	Orexin-A
Uygur, 2016 [60]	Turkey	Caucasian	96	51.4	30.8	27.9	31	50.6	29.6	1.9	Serum	Omentin-1
Zhang, 2018 [62]	China	Asian	30	40.7	28.8	61.5	20	36.1	27.6	1.9	Plasma	Omentin-1
Bİçer, 2021 [56]	Turkey	Caucasian	75	44.2	29.4	≥5	75	34.1	25.9	<5	Plasma	Omentin-1
Oztürk, 2022 [58]	Turkey	Caucasian	210	46.4	32.6	31.6	62	42.2	30.3	2.8	Serum	Orexin-A
Zhu, 2022 [63]	China	Asian	78	56.6	26.9	25.8	28	55.6	26.2	3.4	Plasma	Orexin-A

BMI: Body mass index. AHI: Apnea–hypopnea index. Please note that Liao et al. [43] reported two independent studies.

**Table 2 life-13-00245-t002:** The Joanna Briggs Institute (JBI) critical appraisal checklist for case–control studies.

First Author, Publication Year	The Joanna Briggs Institute (JBI) Critical Appraisal Checklist										Quality (Total Quality Score)
	Q1	Q2	Q3	Q4	Q5	Q6	Q7	Q8	Q9	Q10	
Igarashi, 2003 [57]	No	Yes	Yes	Yes	Yes	No	No	Yes	Yes	Yes	Moderate (7)
Busquets, 2004 [42]	No	No	Yes	Yes	Yes	Yes	Yes	Yes	Yes	Yes	Moderate (7)
Liao, 2005 (i) [43]	No	No	Yes	Yes	Yes	Yes	No	Yes	Yes	Yes	Moderate (6)
Liao, 2005 (ii) [43]	No	No	Yes	Yes	Yes	Yes	No	Yes	Yes	Yes	Moderate (6)
Yao, 2006 [61]	No	Yes	Yes	Yes	Yes	Yes	No	Yes	Yes	Yes	Moderate (7)
Wang, 2013 [41]	Yes	Yes	Yes	Yes	Yes	Yes	No	Yes	Yes	Yes	High (9)
Zirlik, 2013 [64]	Yes	No	Yes	Yes	Yes	Yes	Yes	Yes	Yes	Yes	High (9)
Kurt, 2014 [40]	Yes	No	Yes	Yes	Yes	Yes	Yes	Yes	Yes	Yes	High (9)
Strueva, 2015 [59]	Un	No	No	Yes	Un	Un	Un	Yes	Yes	Yes	Low (4)
Uygur, 2016 [60]	No	Yes	Yes	Yes	Yes	Yes	Yes	Yes	Yes	Yes	High (9)
Zhang, 2018 [62]	Yes	Yes	Yes	Yes	Yes	Yes	No	Yes	Yes	Yes	High (9)
Bİçer, 2021 [56]	Yes	No	Yes	Yes	Yes	No	No	Yes	Yes	Yes	Moderate (7)
Oztürk, 2022 [58]	Yes	No	Yes	Yes	Yes	Yes	No	Yes	Yes	Yes	Moderate (8)
Zhu, 2022 [63]	Yes	Yes	Yes	Yes	Yes	Yes	No	Yes	Yes	Yes	High (10)

Low: score of 1–4, Moderate: score of 5–7, High: score of 8–10. Un: Unclear.

**Table 3 life-13-00245-t003:** Subgroup analysis of the correlation between blood levels of omentin-1 and several variables.

Variable	Subgroup (N)	SMD	95%CI		*p*-Value	I^2^, %
			Min.	Max.		
Ethnicity						
	Caucasian (4)	−0.29	−1.42	0.84	0.62	96
	Asian (2)	−1.95	−4.54	0.64	0.14	98
Sample						
	Serum (3)	−1.66	−3.83	0.51	0.13	99
	Plasma (3)	−0.08	−0.68	0.52	0.79	70
Sample size						
	≥100 (3)	−1.82	−3.85	0.21	0.08	99
	<100 (3)	0.13	−0.64	0.91	0.74	79
Mean BMI of aOSA, kg/m^2^						
	≥30 (3)	−0.33	−2.17	1.51	0.73	97
	<30 (3)	−1.34	−3.52	0.85	0.23	99
Mean BMI of controls, kg/m^2^						
	≥30 (0)	-	-	-	-	-
	<30 (6)	−0.84	−2.19	0.52	0.23	98
Mean age of aOSA, year						
	≥45 (4)	−1.07	−3.01	0.88	0.28	98
	<45 (2)	−0.31	−0.79	0.17	0.20	54
Mean age of controls, year						
	≥45 (3)	−1.57	−3.36	0.23	0.09	97
	<45 (3)	−0.10	−0.58	0.38	0.68	72
Mean AHI of aOSA, events/h						
	≥40 (2)	0.04	−1.34	1.42	0.96	85
	<40 (2)	−2.68	−3.85	−1.52	<0.00001	94
Quality						
	Moderate (5)	−0.98	−2.59	0.62	0.23	98
	High (1)	−0.12	−0.44	0.20	0.46	-

SMD: Std. mean difference. CI: Confidence interval. BMI: Body mass index. AHI: Apnea–hypopnea index. Adults with OSA: aOSA.

**Table 4 life-13-00245-t004:** Subgroup analysis of the correlation between blood levels of orexin-A and several variables.

Variable	Subgroup (N)	SMD	95%CI		*p*-Value	I^2^, %
			Min.	Max.		
Ethnicity						
	Caucasian (3)	−1.14	−3.58	1.29	0.36	98
	Asian (5)	0.30	−0.90	1.51	0.62	96
Sample						
	Serum (1)	0.97	0.67	1.26	<0.00001	-
	Plasma (7)	−0.39	−1.54	0.76	0.51	96
Sample size						
	≥100 (2)	−0.49	−3.36	2.38	0.74	99
	<100 (4)	−0.10	−1.15	0.95	0.85	94
Mean BMI of aOSA, kg/m^2^						
	≥30 (6)	−0.02	−0.98	0.94	0.97	94
	<30 (2)	−0.67	−3.21	1.87	0.60	98
Mean BMI of controls, kg/m^2^						
	≥30 (3)	0.89	0.66	1.12	<0.00001	41
	<30 (5)	−0.88	2.45	0.69	0.27	96
Mean age of aOSA, year						
	≥45 (7)	−0.11	−1.16	0.93	0.83	96
	<45 (1)	−0.85	−1.59	−0.10	0.03	-
Mean age of controls, year						
	≥45 (6)	−0.32	−1.62	0.99	0.64	96
	<45 (2)	0.10	−1.68	1.87	0.92	95
Mean AHI of aOSA, events/h						
	≥40 (2)	−1.58	−5.65	2.49	0.45	98
	<40 (2)	−0.12	−1.93	1.70	0.90	98
Quality						
	Moderate (6)	0.25	−0.59	1.08	0.56	93
	High (1)	−1.96	−2.74	−1.45	<0.00001	**-**
	Low (1)	−0.85	−1.59	−0.10	0.03	**-**

SMD: Std. mean difference. CI: Confidence interval. BMI: Body mass index. AHI: Apnea–hypopnea index. Adults with OSA: aOSA.

**Table 5 life-13-00245-t005:** Meta-regression analysis (mixed-effect model) of the correlation between blood levels of omentin-1 and orexin-A and several variables in the case–control studies in the meta-analysis.

Omentin-1						
Variable	Point Estimate	Standard Error	Lower Limit	Upper Limit	Z-Value	*p*-Value
Publication year	0.08398	0.24376	−0.39379	0.56175	0.34450	0.73047
Sample size	−0.01156	0.00406	−0.01952	−0.00359	−2.84445	**0.00445**
Mean BMI of aOSA	0.62114	0.33477	−0.03499	1.27727	1.85545	0.06353
Mean BMI of controls	−0.62930	0.57486	−1.75600	0.49740	−1.09471	0.27364
Mean age of aOSA	0.03104	0.12756	−0.21897	0.28106	0.24335	0.80773
Mean age of controls	−0.04958	0.08698	−0.22006	0.12090	−0.057001	0.56867
Mean AHI of aOSA	0.06554	0.04650	−0.02560	0.15668	1.40940	0.15872
Quality score	−0.42984	0.99973	−2.38927	1.52959	−0.42996	0.66722
Orexin-A						
Variable	Point estimate	Standard error	Lower limit	Upper limit	Z-value	*p*-value
Publication year	−0.02853	0.07419	−0.17393	0.11687	−0.38457	0.70056
Sample size	0.00559	0.00772	−0.00954	0.02072	0.72432	0.46887
Mean BMI of aOSA	0.01832	0.14761	−0.27099	0.30763	0.12411	0.90123
Mean BMI of controls	0.17726	0.13499	−0.08731	0.44183	1.31314	0.18914
Mean age of aOSA	−0.07771	0.10418	−0.28190	0.12647	−0.74595	0.45570
Mean age of controls	−0.03628	0.10193	−0.23607	0.16351	−0.35589	0.72192
Mean AHI of aOSA	−0.32946	0.60582	−151685	0.85793	−0.54382	0.58657
Quality score	−0.24502	0.31513	−0.82267	0.37262	−0.77753	0.43685

BMI: Body mass index. AHI: Apnea–hypopnea index. Bold number means statistically significant (*p* < 0.05). Adults with OSA: aOSA.

## Data Availability

No new data were created or analyzed in this study. Data sharing is not applicable to this article.

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
