# Peer review of "Evaluation of Blood Levels of Omentin-1 and Orexin-A in Adults with Obstructive Sleep Apnea: A Systematic Review and Meta-Analysis"

_life, 2023, doi:10.3390/life13010245_

Round 1

Reviewer 1 Report

The authors proposed an interesting systematic review & meta-analysis pertaining to the evaluation of blood levels of omentin-1 and orexin-A in adults with obstructive sleep apnea. Before my final decision, the following issues should be addressed:

Major:
- 95%CI should be stated as an interval ("two numbers"), not one, see line 34
- Why was the literature screened only by one author?
- Did you choose any inclusion/exclusion criteria pertaining to the control group selection? Based on table 1: it is quite heterogeneous: nonOSA vs AHI>30, BMI: normal, overweight, obese... It may affect further results...
- Was data collection checked by any other authors? What was done in the case of any doubts?
- Line 151-2: "Some studies reported median and 25th–75th percentiles [54] and median (min-max) [55] that we estimated to mean (standard deviation)." The median with interquartile range (1.-3. quartile) is reliable in the case of both normal and other distribution of data, when mean +- sd is reliable just for data with normal distribution. Therefore the authors should better explain their decision.
-  Why is Liao, 2005 treated like two publications, but it has one ref [43] in table 2?

Minor:
- Line 27-29: "Four well-known, well-established and important databases were systematically searched until November 14, 2022, without any restrictions." - I suggest replacing the phrase "Four well-known, well-established and important databases were" by "Four databases (names) ..." or "The following databases ... were...". It will be more informative
- Line 31: The authors should provide company and country of the manufacturer of Review Manager 5.3 software
- Line 33: there is no need to introduce abbreviation (SMDs) that was not used further in abstract
- line 71: extra spaces after first "-" and before second "-" are needed
- OM-1 - Why did you change the font (size?)?
- Figure 1: "other database" means.. Google scholar? Or anything more?
- Funding information: please add information about APC founding (Stat funds?).

Author Response

We thank Reviewer #1 for their care devoted to the present manuscript; the comments and suggestions helped us to improve the quality of the revision. Thank you once again!

Reviewer 2 Report

The study Evaluation of blood levels of omentin-1 and orexin-A in adults 2 with obstructive sleep apnea: A systematic review and meta-3 analysis 4 is interesting. It gives more evidence about the association between OXA, OM-1, and OSA. It was written systematically, and the method was described clearly including the statistics part. The authors also follow all parts of the PRISMA 2020.
I have no suggestion for changing any part of the manuscript.

Author Response

We thank Reviewer #2 for the care devoted to the present manuscript. The comments were very encouraging and motivating to undertake a thorough revision. Thank you once again!

Round 2

Reviewer 1 Report

Thank you very much for addressing all issues. Your paper can be published in its current form.

Congratulation!